# Regularized Asymptotic Solutions of a Singularly Perturbed Fredholm Equation with a Rapidly Varying Kernel and a Rapidly Oscillating Inhomogeneity

**Dana Bibulova [1], Burkhan Kalimbetov [2],\* and Valeriy Safonov [3]**

[1]    Department of Higher Mathematics, South Kazakhstan University Named after M. Auezov, Tauke-Khan Ave., 5, Shymkent 160000, Kazakhstan; danass86@mail.ru

[2]    Department of Mathematics, Akhmed Yassawi University, B. Sattarkhanov 29, Turkestan 161200, Kazakhstan

[3]    Department of Higher Mathematics, National Research University «MPEI», Krasnokazarmennaya 14, 111250 Moscow, Russia; singsaf@yandex.ru

\*    Correspondence: burkhan.kalimbetov@ayu.edu.kz

**Abstract:** This article investigates an equation with a rapidly oscillating inhomogeneity and with a rapidly decreasing kernel of an integral operator of Fredholm type. Earlier, differential problems of this type were studied in which the integral term was either absent or had the form of a Volterra-type integral. The presence of an integral operator and its type significantly affect the development of an algorithm for asymptotic solutions, in the implementation of which it is necessary to take into account essential singularities generated by the rapidly decreasing kernel of the integral operator. It is shown in tise work that when passing the structure of essentially singular singularities changes from an integral operator of Volterra type to an operator of Fredholm type. If in the case of the Volterra operator they change with a change in the independent variable, then the singularities generated by the kernel of the integral Fredholm-type operators are constant and depend only on a small parameter. All these effects, as well as the effects introduced by the rapidly oscillating inhomogeneity, are necessary to take into account when developing an algorithm for constructing asymptotic solutions to the original problem, which is implemented in this work.

**Keywords:** singular perturbation; integro-differential equation; rapidly oscillating inhomogeneity; regularization; asymptotic convergence

**MSC:** 34K26; 45J05

## 1. Introduction

Integro-differential equations

$$
L_\varepsilon y(t,\varepsilon) \equiv \varepsilon \frac{dy}{dt} - a(t)y - \int_0^\alpha e^{\frac{1}{\varepsilon}\int_s^\alpha \mu(\theta)d\theta} K(t,s)y(s,\varepsilon)ds =
$$
$$
= h_1(t) + h_2(t)e^{\frac{i\beta(t)}{\varepsilon}}, \ \ y(0,\varepsilon) = y^0, \ \ t \in [0,T] \tag{1}
$$

with rapidly changing Volterra-type kernels ($\alpha = t$) have been studied from various positions in a number of works (see, for example, [1] and its bibliography). The problems were considered on the construction of a regularized asymptotics for the solution of problem type (1) in the case of stability of the operator $a(t)$ and the spectral value $\mu(t)$ of the kernel of the integral operator [2–7]. As for the integro-differential equations (1) with rapidly changing kernels of the Fredholm type ($\alpha = T$), it was assumed that the results obtained for the Volterra equations are automatically extended to equations of the Fredholm type. However, when considering the simplest case of scalar Equation (1) for (see, for example, [8–14]) it turned out that the spectral value does not participate in the

regularization of problem (1) (in contrast to the case $\alpha = t$), and the coefficients of the elements of the space of resonance-free solutions (in the terminology of S.A. Lomov [15–17] depend on the exponentials

$$\sigma_1 = \exp\{\tfrac{1}{\varepsilon}\int_0^1 a(\theta)d\theta\}, \sigma_2 = \exp\{\tfrac{i}{\varepsilon}\int_0^1 \beta'(\theta)d\theta\}, \sigma_3 = \exp\{\tfrac{1}{\varepsilon}\int_0^1 \mu(\theta)d\theta\},$$

$$\sigma_4 = \exp\{\tfrac{i}{\varepsilon}\beta(0)\}, (\beta'(t) > 0 \ \forall t \in [0,1])$$

bounded for $\varepsilon \to +0$, if $a(t) < 0$ and $\mu(t) < 0, \beta'(t) > 0$ ($t \in [0,1]$), $\beta(t)$ is a real function). Therefore, the regularization of problem (1) and the theory of normal and unique solvability of the corresponding iterative problems do not fit into the previously developed scheme for equations of the Volterra type and should be revised, taking into account those changes which are introduced by the Fredholm operator. Fredholm-type integro-differential equations with slow and rapidly changing kernels have been studied in [18]. In recent years, the main attention of researchers has been focused on the development of asymptotic solutions for integro-differential equations with rapidly oscillating coefficients in the presence of rapidly oscillating inhomogeneities [19]. Therefore, in this paper, an attempt is made to create an algorithm for constructing an asymptotic solution of problem (1) at $\varepsilon \to +0$. Without a loss of generality, we can assume that $\alpha = T = 1$, and we will proceed to the study of the problem (1).

Thus, in this paper we consider the following Cauchy problem:

$$L_\varepsilon y(t,\varepsilon) \equiv \varepsilon\frac{dy}{dt} - a(t)y - \int_0^1 e^{\frac{1}{\varepsilon}\int_s^1 \mu(\theta)d\theta}K(t,s)y(s,\varepsilon)ds =$$

$$= h_1(t) + h_2(t)e^{\frac{i\beta(t)}{\varepsilon}}, y(0,\varepsilon) = y^0, t \in [0,1] \tag{2}$$

with the Fredholm type of integral operator.

## 2. Regularization of the Problem (1)

The problem (1) will be considered under the following conditions:

$(1)\, a(t), \mu(t), \beta(t) \in C^\infty([0,1],\mathbb{R}), h_1(t), h_2(t) \in ([0,1],\mathbb{C}), K(t,s) \in C^\infty(\{0 \le s \le t \le 1\},\mathbb{C});$

$(2)\, a(t) \ne \mu(t), \mu(t) < 0, a(t) < 0\, \forall t \in [0,1]$.

Let us denote $\lambda_1(t) \equiv a(t), \lambda_2(t) \equiv i\beta'(t), \lambda_3(t) \equiv \mu(t)$ and call the set $\{\lambda_j(t)\}$ the spectrum of problem (1). We introduce the regularizing variables

$$\tau_j = \frac{1}{\varepsilon}\int_0^t \lambda_j(\theta)d\theta = \frac{\psi_j(t)}{\varepsilon}, j = 1,2$$

along the points of the spectrum $\lambda_1(t)$ and $\lambda_2(t)$ of the problem (1) (in this case, as will be shown below, the variable $\tau_3 = \varepsilon^{-1}\int_0^t \lambda_3(\theta)d\theta$ does not participate in the regularization). For the "extension" $\tilde{y}(t,\tau,\varepsilon)$ we obtain the following problem:

$$L_\varepsilon\tilde{y}(t,\tau,\varepsilon) \equiv \varepsilon\frac{\partial\tilde{y}}{\partial t} + \sum_{j=1}^2 \lambda_j(t)\frac{\partial\tilde{y}}{\partial\tau_j} - \lambda_1(t)\tilde{y} - \int_0^1 e^{\frac{1}{\varepsilon}\int_s^1 \lambda_3(\theta)d\theta}K(t,s)\tilde{y}(s,\tfrac{\psi(s)}{\varepsilon},\varepsilon)ds =$$

$$= h_1(t) + h_2(t)e^{\tau_2}\sigma_4, \quad \tilde{y}(0,0,\varepsilon) = y^0, \quad t \in [0,1] \tag{3}$$

where $\tau = (\tau_1, \tau_2), \psi = (\psi_1, \psi_2)$. The function $\tilde{y}(t,\tau,\varepsilon)$ satisfies the necessary regularization condition: $\tilde{y}(t,\tfrac{\psi(t)}{\varepsilon},\varepsilon) \equiv y(t,\varepsilon)(y(t,\varepsilon)$ is the exact solution to problem (1)). However, problem (3) cannot be considered completely regularized, since the integral term

$$J\tilde{y} \equiv J\Big(\tilde{y}(t,\tau,\varepsilon)|_{t=s,\tau=\psi(s)/\varepsilon}\Big) = \int_0^1 e^{\frac{1}{\varepsilon}\int_s^1 \lambda_3(\theta)d\theta}K(t,s)\tilde{y}(s,\tfrac{\psi(s)}{\varepsilon},\varepsilon)ds. \tag{4}$$

has not been regularized in it. For its regularization, as is known, it is necessary to introduce a class $M_\varepsilon$, that is asymptotically invariant with respect to the operator $J$ (see [15], pp. 62–64).

**Definition 1.** *We say that a vector function $y(t, \tau, \sigma)$ belongs to the space $U$, if it is represented by a sum of the form*

$$y(t, \tau) \equiv y(t, \tau, \sigma) = y_0(t, \sigma) + \sum_{j=1}^{2} y_j(t, \sigma) e^{\tau_j} \tag{5}$$

*where the functions $y_j(t, \sigma)$ are polynomials in $\sigma = (\sigma_1, \ldots, \sigma_4)$ with coefficients from the class $C^\infty([0,1], \mathbb{C})$, i.e.,*

$$y_j(t, \sigma) = \sum_{|m|=0}^{N_j} y_j^{(m_1, \ldots, m_4)}(t) \sigma_1^{m_1} \cdots \sigma_4^{m_4},$$

$$y_j^{(m_1, \ldots, m_4)}(t) \in C^\infty([0,1], \mathbb{C}), \quad 0 \le |m| \equiv m_1 + \cdots + m_4, \ N_i < \infty, \ j = 0, 1, 2.$$

*We take as $M_\varepsilon$ the class $U|_{\tau = \frac{\psi(t)}{\varepsilon}}$. It should be shown that the image $Jy(t, \tau)$ on functions (4) can be represented in the form of a series*

$$\sum_{k=0}^{\infty} \varepsilon^k \left( \sum_{j=1}^{2} z_j^{(k)}(t, \sigma) e^{\tau_j} + z_0^{(k)}(t, \sigma) \right) \Big|_{\tau = \frac{\psi(t)}{\varepsilon}},$$

*converging asymptotically to $Jy$ at $\varepsilon \to +0$ (uniformly with respect to $t \in [0, 1]$). Substituting (5) in, we will have*

$$Jy(t, \tau, \sigma) = \int_0^1 K(t, s) y_0(s, \sigma) e^{\frac{1}{\varepsilon} \int_s^1 \lambda_3(\theta) d\theta} ds + \sum_{j=1}^{2} \int_0^1 K(t, s) y_j(s, \sigma) e^{\frac{1}{\varepsilon} \int_s^1 \lambda_3(\theta) d\theta + \frac{1}{\varepsilon} \int_0^s \lambda_j(\theta) d\theta} ds.$$

*We take the integrals here by parts:*

$$J_0(t, \varepsilon) = \int_0^1 e^{\frac{1}{\varepsilon} \int_s^1 \lambda_3(\theta) d\theta} K(t, s) y_0(s, \sigma) ds = \varepsilon \int_0^1 \frac{K(t, s) y_0(s, \sigma)}{-\lambda_3(s)} d\left(\exp\left(\frac{1}{\varepsilon} \int_s^1 \lambda_3(\theta) d\theta\right)\right) =$$

$$= \varepsilon \left[ \frac{K(t, 1) y_0(1, \sigma)}{-\lambda_3(1)} - \frac{K(t, 0) y_0(0, \sigma)}{-\lambda_3(0)} \sigma_3 \right] - \varepsilon \int_0^1 \exp\left(\frac{1}{\varepsilon} \int_s^1 \lambda_3(\theta) d\theta\right) \frac{\partial}{\partial s} \left(\frac{K(t, s) y_0(s, \sigma)}{-\lambda_3(s)}\right) ds =$$

$$= \sum_{\nu=0}^{\infty} (-1)^\nu \varepsilon^{\nu+1} [(I_0^\nu (K(t, s) y_0(s, \sigma)))_{s=1} - (I_0^\nu (K(t, s) y_0(s, \sigma)))_{s=0} \sigma_3], \tag{5a}$$

$$J_j(t, \varepsilon) = \int_0^1 \exp\left(\frac{1}{\varepsilon} \int_s^1 \lambda_3(\theta) d\theta + \frac{1}{\varepsilon} \int_0^s \lambda_j(\theta) d\theta\right) K(t, s) y_j(s, \sigma) ds \equiv$$

$$\equiv \sigma_3 \int_0^1 \exp\left(\frac{1}{\varepsilon} \int_0^s (\lambda_j(\theta) - \lambda_3(\theta)) d\theta\right) K(t, s) y_j(s, \sigma) ds \equiv$$

$$\equiv \sigma_3 \varepsilon \int_0^1 \frac{K(t, s) y_j(s, \sigma)}{\lambda_j(s) - \lambda_3(s)} d\left(\exp\left(\frac{1}{\varepsilon} \int_0^s (\lambda_j(\theta) - \lambda_3(\theta)) d\theta\right)\right) =$$

$$= \varepsilon \sigma_3 \left[ \frac{K(t, 1) y_j(1, \sigma)}{\lambda_j(1) - \lambda_3(1)} \frac{\sigma_j}{\sigma_3} - \frac{K(t, 0) y_j(0, \sigma)}{(\lambda_j(0) - \lambda_3(0))} \right] -$$

$$- \varepsilon \sigma_3 \int_0^1 \exp\left(\frac{1}{\varepsilon} \int_0^s (\lambda_j(\theta) - \lambda_3(\theta)) d\theta\right) \frac{\partial}{\partial s} \left(\frac{K(t, s) y_j(s, \sigma)}{\lambda_j(s) - \lambda_3(s)}\right) ds =$$

$$= \sum_{\nu=0}^{\infty} (-1)^{\nu} \varepsilon^{\nu+1} \left[ (I_j^{\nu}(K(t,s)y_j(s,\sigma)))_{s=1}\sigma_j - (I_j^{\nu}(K(t,s)y_j(s,\sigma)))_{s=0}\sigma_3 \right] \quad (5b)$$

*where* $j = 1, 2$ *and the operators are introduced:*

$$I_0^0 = \frac{1}{-\lambda_3(s)}, I_0^{\nu} = \frac{1}{-\lambda_3(s)} \frac{\partial}{\partial s} I_0^{\nu-1}, (\nu \geq 1),$$

$$I_j^0 = \frac{1}{\lambda_j(s) - \lambda_3(s)}, I_j^{\nu} = \frac{1}{\lambda_j(s) - \lambda_3(s)} \frac{\partial}{\partial s} I_j^{\nu-1}, (\nu \geq 1, j = 1, 2). \quad (5c)$$

*It is easy to show (see, for example, [20], pp. 291–294) that the series (5a, b) converge asymptotically (at* $\varepsilon \to +0$*) to the corresponding integrals* $J_j(t, \varepsilon)$ *(uniformly with respect to* $t \in [0,1]$*), and hence the image* $Jy(t, \tau)$ *is represented as series* $\sum_{k=0}^{\infty} \varepsilon^k (\sum_{j=1}^{2} z_j^{(k)}(t,\sigma)e^{\tau_j} + z_0^{(k)}(t,\sigma)g)|_{\tau=\frac{\psi(t)}{\varepsilon}}$*, also converging asymptotically to* $Jy(t, \tau)$ *uniformly with respect to* $t \in [0,1]$*). Thus, it is shown that the class* $M_{\varepsilon} = U|_{\tau=\frac{\psi(t)}{\varepsilon}}$ *is asymptotically invariant with respect to the operator* $J$*.*

*Now let* $\tilde{y}(t, \tau, \varepsilon)$ *be an arbitrary continuous in* $(t, \tau) \in [0,1] \times \Pi (\Pi = \{\tau : \text{Re}\tau_j \leq 0, j = 1, 2\}$ *function represented by the series*

$$\tilde{y}(t, \tau, \varepsilon) = \sum_{k=0}^{\infty} \varepsilon^k y_k(t, \tau, \sigma), y_k(t, \tau, \sigma) \in U \quad (6)$$

*converging asymptotically at* $\varepsilon \to +0$ *(uniformly with respect to* $t \in [0,1]$*). Substituting (6) into (4) and collecting the coefficients at the same degrees of* $\varepsilon$*, we obtain the series*

$$J\tilde{y}(t, \tau, \varepsilon) = \sum_{k=0}^{\infty} \varepsilon^k Jy_k(t, \tau, \sigma) = \sum_{r=0}^{\infty} \varepsilon^r \sum_{s=0}^{r} R_{r-s}y_s(t, \tau, \sigma)$$

*converging asymptotically to* $J\tilde{y}$ *for* $\varepsilon \to +0$ *(uniformly in* $t \in [0,1]$*). Here* $R_{\nu} : U \to U$ *(the operators of order in* $\varepsilon$*) are of the following form:*

$$R_0 y(t, \tau, \sigma) \equiv 0,$$

$$R_1(t, \tau, \sigma) = \left[ \frac{K(t,1)y_0(1,\sigma)}{-\lambda_3(1)} - \frac{K(t,0)y_0(0,\sigma)}{-\lambda_3(0)}\sigma_3 \right] +$$

$$+ \sum_{j=1}^{2} \left[ \frac{K(t,1)y_j(1,\sigma)}{\lambda_j(1) - \lambda_3(1)}\sigma_j - \frac{K(t,0)y_j(0,\sigma)}{\lambda_j(0) - \lambda_3(0)}\sigma_3 \right], \quad (6a)$$

$$R_{\nu+1} y(t, \tau, \sigma) = (-1)^{\nu}[(I_0^{\nu}(K(t,s)y_0(s,\sigma)))_{s=1} - (I_0^{\nu}(K(t,s)y_0(s,\sigma)))_{s=0}\sigma_3] +$$

$$+ \sum_{j=1}^{2} (-1)^{\nu} \left[ (I_j^{\nu}(K(t,s)y_j(s,\sigma)))_{s=1} - (I_j^{\nu}(K(t,s)y_j(s,\sigma)))_{s=0}\sigma_3 \right] \quad (6b)$$

*where* $I_j^{\nu}$ *are the operators (5c),* $j = \overline{0,2}, \nu \geq 0$*, introduced above, and* $y(t, \tau, \sigma)$ *is the function (5).*

**Definition 2.** *By a formal extension of an operator* $J$*, we mean an operator* $\tilde{J}$*, acting on any continuous in* $(t, \tau) \in [0,1] \times \Pi$ *function* $\tilde{y}(t, \tau, \varepsilon)$ *of the form (6) according to the law*

$$\tilde{J}\tilde{y} \equiv \tilde{J}\left( \sum_{k=0}^{\infty} \varepsilon^k y_k(t, \tau, \sigma) \right) = \sum_{r=0}^{\infty} \varepsilon^r \sum_{s=0}^{r} R_{r-s}y_s(t, \tau, \sigma). \quad (7)$$

*Equality (7) is the basis for the definition of the operator* $\tilde{J}$*, extended with respect to the integral operator* $J$*. Despite the fact that the extension* $\tilde{J}$ *of the operator* $J$ *is defined formally, it is quite*

*possible to use it (see Theorem 3 below) when constructing an asymptotic solution of finite order in ε. Now we can write the problem completely regularized with respect to the original (1):*

$$L_\varepsilon \tilde{y}(t, \tau, \sigma, \varepsilon) \equiv \varepsilon \frac{\partial \tilde{y}}{\partial t} + \sum_{j=1}^{2} \lambda_j(t) \frac{\partial \tilde{y}}{\partial \tau_j} - \lambda_1(t)\tilde{y} - \tilde{J}\tilde{y} =$$

$$= h_1(t) + h_2(t)e^{\tau_2}\sigma_4, \tilde{y}(t, \tau, \varepsilon)|_{t=0, \tau=0} = y^0 \tag{8}$$

*where the operator $\tilde{J}$ has the form (7).*

### 3. Iterative Problems and Their Solvability in the Space *U*

Substituting (6) into (8) and equating the coefficients at the same degrees of ε, we obtain the following iterative problems:

$$Ly_0(t, \tau, \sigma) \equiv \sum_{j=1}^{2} \lambda_j(t) \frac{\partial y_0}{\partial \tau_j} - \lambda_1(t)y_0 =$$
$$= h_1(t) + h_2(t)e^{\tau_2}\sigma_4, y_0(0,0) = y^0; \tag{$9_0$}$$

$$Ly_1(t, \tau, \sigma) = -\frac{\partial y_0}{\partial t} + R_1 y_0, y_1(0,0) = 0; \tag{$9_1$}$$

$$Ly_2(t, \tau, \sigma) = -\frac{\partial y_1}{\partial t} + R_1 y_1 + R_2 y_0, y_2(0,0) = 0; \tag{$9_2$}$$

$$\ldots$$

$$Ly_k(t, \tau, \sigma) = -\frac{\partial y_{k-1}}{\partial t} + R_k y_0 + \ldots + R_1 y_{k-1}, \ y_k(0,0) = 0, k \geq 1. \tag{$9_k$}$$

Each of the iterative problems $(9_k)$ has the form

$$Ly(t, \tau, \sigma) \equiv \sum_{j=1}^{2} \lambda_j(t) \frac{\partial y}{\partial \tau_j} - \lambda_1(t)y = H(t, \tau, \sigma), y(0,0,\sigma) = y_* \tag{10}$$

where $H(t, \tau, \sigma) = H_0(t, \sigma) + \sum_{j=1}^{2} H_j(t, \sigma)e^{\tau_j}$. We introduce in the space *U* a scalar product (for each $t \in [0, 1]$ and $\sigma$) :

$$< y(t, \tau, \sigma), z(t, \tau, \sigma) > \equiv < \sum_{j=1}^{2} y_j(t, \sigma)e^{\tau_j} + y_0(t, \sigma), \sum_{j=1}^{2} z_j(t, \sigma)e^{\tau_j} +$$

$$+ z_0(t, \sigma) > \stackrel{def}{=} \sum_{j=0}^{2} (y_j(t, \sigma), z_j(t, \sigma))$$

where $(*, *)$ is the usual scalar product in $\mathbb{C}$. Let us prove the following statement.

**Theorem 1.** *Let $H(t, \tau) \in U$, and conditions (1) and (2) be satisfied. Then, for the solvability of Equation (10) in the space U , it is necessary and sufficient that*

$$< H_1(t, \tau, \sigma), e^{\tau_1} > \equiv 0 \Leftrightarrow H_1(t, \sigma) \equiv 0, \forall t \in [0, 1]. \tag{11}$$

**Proof.** Defining the solution of the Equation (10) in the form of function (5), we obtain the identity

$$\sum_{j=1}^{2} [\lambda_j(t) - \lambda_1(t)] y_j(t, \sigma)e^{\tau_j} - \lambda_1(t)y_0(t, \sigma) = H_0(t, \sigma) + \sum_{j=1}^{2} H_j(t, \sigma)e^{\tau_j}.$$

Equating here separately the free terms and the coefficients at the exponentials $e^{\tau_j}$, we will have

$$-\lambda_1(t)y_0(t,\sigma) = H_0(t,\sigma), \tag{$12_0$}$$

$$[\lambda_j(t) - \lambda_1(t)]y_j(t,\sigma) = H_j(t,\sigma), j = 1,2. \tag{$12_j$}$$

Since $\lambda_1(t) \neq 0 \forall t \in [0,1]$, that the Equation ($12_0$) has a unique solution

$$y_0(t,\sigma) = -\frac{H_0(t,\sigma)}{\lambda_1(t,\sigma)}. \tag{13}$$

Since $\lambda_1(t)$ is a real function and $\lambda_2(t) = i\beta'(t)$ is purely imaginary, the Equation ($12_2$) has a unique solution in the space $C^\infty([0,1],\mathbb{C})$. For the solvability of the Equation ($12_1$) in the space $C^\infty([0,1],\mathbb{C})$ it is necessary and sufficient for identity (11) to hold. Thus, Theorem 1 is proved. $\square$

**Remark 1.** *It follows from the Equalities ($12_0$)–(13) that under conditions (1) and (2) and condition (11), Equation (10) has the following solution in the space U:*

$$y(t,\tau,\sigma) = y_0(t,\sigma) + \alpha_1(t,\sigma)e^{\tau_1} + y_2(t,\sigma)e^{\tau_2} \tag{14}$$

*where $\alpha_1(t,\sigma) \in C^\infty([0,1],\mathbb{C})$ is an arbitrary function,*

$$y_0(t,\sigma) = -\lambda_1^{-1}(t)H_0(t,\sigma), \quad y_2(t,\sigma) = (\lambda_2(t) - \lambda_1(t))^{-1}H_2(t,\sigma).$$

*Thus, the solution (14) of the Equation (10) is determined ambiguously in the space U. Let now $y_* \in \mathbb{C}$ be a fixed constant vector. Consider the following problem:*

$$y(0,0,\sigma) = y_*, \tag{15}$$

$$< -\frac{\partial y}{\partial t} + R_1 y + Q(t,\tau,\sigma), e^{\tau_1} >\equiv 0, \forall t \in [0,1]$$

*where $Q(t,\tau,\sigma) = Q_0(t,\sigma) + \sum_{j=1}^2 Q_i(t,\sigma)e^{\tau_j}$ is the well-known vector function of the space U, and $R_1$ is the order operator described above (see (6a)). Let us prove the following statement.*

**Theorem 2.** *Let conditions (1) and (2) be satisfied and the vector function $H(t,\tau,\sigma) \in U$ satisfies the orthogonality conditions (11). Then the problem (10) under additional conditions (15) has a unique solution in the space U.*

**Proof.** Since the condition (11) is satisfied, Equation (10) has a solution in the space $U$ in the form of the function (14), where $\alpha_1(t,\sigma) \in C^\infty([0,1],\mathbb{C})$ is an arbitrary function. Submitting (14) to the condition $y(0,0) = y_*$, we have

$$y_* = y_0(0,\sigma) + \alpha_1(0,\sigma) + \frac{H_2(0,\sigma)}{\lambda_2(0) - \lambda_1(0)} \Leftrightarrow$$

$$\Leftrightarrow \alpha_1(0,\sigma) = y_* + \frac{H_0(0,\sigma)}{\lambda_1(0)} - \frac{H_2(0,\sigma)}{\lambda_2(0) - \lambda_1(0)}.$$

Let us now subordinate (14) to the second condition (15):

$$-\frac{\partial y_0}{\partial t} + R_1 y_0 + Q(t,\tau) = -\dot{y}_0(t,\sigma) - \dot{\alpha}_1(t,\sigma)e^{\tau_1} - \dot{y}_2(t,\sigma)e^{\tau_2} +$$

$$+\left[\frac{K(t,1)y_0(1,\sigma)}{-\lambda_3(1)}\sigma_3 - \frac{K(t,0)y_0(0,\sigma)}{-\lambda_3(0)}\right] + \left[\frac{K(t,1)\alpha_1(1,\sigma)}{\lambda_1(1) - \lambda_3(1)}\sigma_1 - \frac{K(t,0)\alpha_1(0,\sigma)}{\lambda_1(0) - \lambda_3(0)}\sigma_3\right] +$$

$$+ \left[ \frac{K(t,1)y_2(1,\sigma)}{\lambda_2(1) - \lambda_3(1)}\sigma_2 - \frac{K(t,0)y_2(0,\sigma)}{\lambda_2(0) - \lambda_3(0)}\sigma_3 \right] + Q(t,\tau).$$

Considering that here the expressions in square brackets do not contain an exponent $e^{\tau_1}$, we perform scalar multiplication in the second equality (15). This gives

$$-\dot{\alpha}_1(t,\sigma) + Q_1(t) = 0 \Leftrightarrow \alpha_1(t,\sigma) = \int_0^t Q_1(\theta)d\theta + y_* + \frac{H_0(0,\sigma)}{\lambda_1(0)} - \frac{H_2(0,\sigma)}{\lambda_2(0) - \lambda_1(0)}$$

and hence, we construct the solution (14) of the problem (10) in the space $U$ in a unique way. Theorem 2 is proved. □

## 4. Construction of the Solution to the First Iterative Problem

Let us apply Theorem 1 to iterative problems $(9_k)$. Since the right-hand side $h_1(t) + h_2(t)e^{\tau_2}\sigma_4$ of the Equation $(9_0)$ satisfies condition (11), the solution $y_0(t,\tau) \in U$ of the first iterative problem $(9_0)$ has the form

$$y_0(t,\tau,\sigma) = \frac{h_1(t)}{-\lambda_1(t)} + \alpha_1^{(0)}(t,\sigma)e^{\tau_1} + \frac{h_2(t)}{\lambda_2(t) - \lambda_1(t)}e^{\tau_2}\sigma_4 \qquad (16)$$

where $\alpha_1^{(0)}(t,\sigma) \in C^\infty[0,1]$ is an arbitrary function. Submitting this solution to the initial condition $y_0(0,0,\sigma) = y^0$, we find

$$\frac{h_1(0)}{-\lambda_1(0)} + \alpha_1^{(0)}(0,\sigma) + \frac{h_2(0)}{\lambda_2(0)-\lambda_1(0)}\sigma_4 = y^0 \Leftrightarrow$$

$$\Leftrightarrow \alpha_1^{(0)}(0,\sigma) = y^0 + \frac{h_1(0)}{\lambda_1(0)} - \frac{h_2(0)}{\lambda_2(0)-\lambda_1(0)}\sigma_4. \qquad (17)$$

For the final calculation of the function $\alpha_1^{(0)}(t,\sigma)$, it is necessary to write down conditions (11) for the next iterative problem $(9_1)$. Since $R_1y_0(t,\tau)$ does not contain an exponent, then, under the orthogonality conditions (11), it can be omitted and an equality can be obtained $\dot{\alpha}_1^{(0)}(t,\sigma) = 0$, which, taking into account the initial condition (17), leads to an unambiguous calculation of the function

$$\alpha_1^{(0)}(t,\sigma) = y^0 + \frac{h_1(0)}{\lambda_1(0)} - \frac{h_2(0)}{\lambda_2(0) - \lambda_1(0)}\sigma_4 = \text{const}$$

and hence to an unambiguous calculation of the solution (16) of the first iterative problem $(9_0)$ in the space $U$.

**Remark 2.** *The solution of the following problem $(9_1)$ is determined from the system*

$$Ly_1(t,\tau,\sigma) = -\frac{\partial y_0}{\partial t} + R_1y_0, y_1(0,0) = 0,$$

$$< -\frac{\partial y_1}{\partial t} + R_1y_1 + R_2y_0, e^{\tau_1} > \equiv 0 \forall t \in [0,1]. \qquad (18)$$

*As in the previous case, the expression $R_1y_1$ and $R_2y_0$ does not contain an exponent $e^{\tau_1}$, therefore, under orthogonality conditions (18), they can be omitted, and then the solution $y_1(t,\tau) \in U$ of the iterative problem $(9_1)$ will be determined from the system*

$$Ly_1(t,\tau,\sigma) = -\frac{\partial y_0}{\partial t} + R_1y_0, y_1(0,0) = 0,$$

$$< -\frac{\partial y_1}{\partial t}, e^{\tau_1} > \equiv 0 \forall t \in [0,1].$$

*The same situation takes place for all subsequent iterative problems $(9_k)(k \geq 2)$. Thus, the influence of the Fredholm-type integral operator in (1) affects only the formation of particular solutions of equations for functions $\alpha_1^{(k)}(t,\sigma)$, while in Volterra systems the kernel $K(t,s)$ of the integral operator participates in the formation of common solutions for these functions.*

### 5. Justification of the Asymptotic Convergence of Formal Solutions to the Exact Solutions

Applying Theorems 1 and 2 to iterative problems $(9_k)$, we can uniquely calculate their solutions $y_k(t,\tau,\sigma)$ in the space $U$. Denote the $N$-th partial sum of series (6) by $S_N(t,\tau,\sigma)$, and through $y_{\varepsilon N}(t) = S_N(t,\frac{\psi(t)}{\varepsilon},\varepsilon)$ is the restriction of this sum at $\tau = \frac{\psi(t)}{\varepsilon}$. It is easy to prove the following assertion (see, for example, [15], pp. 37–40).

**Lemma 1.** *Let conditions (1) and (2) be satisfied. Then the function $y_{\varepsilon N}(t)$ is a formal asymptotic solution of the problem (1) of order N, that is, it satisfies the problem*

$$\varepsilon\frac{dy_{\varepsilon N}}{dt} - a(t)y_{\varepsilon N} - \int_0^1 \exp\left(\frac{1}{\varepsilon}\int_s^1 \mu(\theta)d\theta\right)K(t,s)y_{\varepsilon N}(s)ds =$$

$$= h_1(t) + h_2(t)e^{\frac{i\beta(t)}{\varepsilon}} + \varepsilon^{N+1}F_N(t,\varepsilon), y_{\varepsilon N}(0) = y^0 \tag{19}$$

*where $||F_N(t,\varepsilon)||_{C[0,1]} \leq \bar{F}(\bar{F} > 0$ is a constant independent of $\varepsilon$ at $\varepsilon \in (0,\varepsilon_0], \varepsilon_0$ is small enough).*

To prove the Theorem on the estimate of the remainder term, we first consider the integro-differential equation

$$\varepsilon\frac{dz}{dt} = a(t)z + \int_0^1 \exp\left(\frac{1}{\varepsilon}\int_s^1 \mu(\theta)d\theta\right)K(t,s)z(s,\varepsilon)ds + H(t,\varepsilon), z(0,\varepsilon) = 0 \tag{$20_0$}$$

*and try to estimate the norm of its solution $z(t,\varepsilon)$ in terms of the norm of the right-hand side $H(t,\varepsilon)$. The function $Y(t,s,\varepsilon) = e^{\frac{1}{\varepsilon}\int_s^t a(\theta)d\theta}$ is the fundamental Cauchy solution for a homogeneous equation $\varepsilon\dot{z} = a(t)z$. Under conditions (1) and (2) it is uniformly bounded, i.e., $||Y(t,s,\varepsilon)|| \leq c_0 = \text{const}$ for all $(t,s,\varepsilon) \in \{0 \leq s \leq t \leq 1, \varepsilon > 0\}$. Let us convert the Equation $(20_0)$, using $Y(t,s,\varepsilon)$; we obtain the equivalent integral equation*

$$z(t,\varepsilon) = \frac{1}{\varepsilon}\int_0^t e^{\frac{1}{\varepsilon}\int_x^t a(\theta)d\theta}\left(\int_0^1 \left(e^{\frac{1}{\varepsilon}\int_s^1 \mu(\theta)d\theta}\right)K(x,s)z(s,\varepsilon)ds\right)dx + \frac{1}{\varepsilon}\int_0^t e^{\frac{1}{\varepsilon}\int_x^t a(\theta)d\theta}H(x,\varepsilon)dx.$$

*Denoting $H_1(t,\varepsilon) \equiv \int_0^t e^{\frac{1}{\varepsilon}\int_x^t a(\theta)d\theta}H(x,\varepsilon)dx$ and changing the order of integration in the iterated integral, we obtain the following integral equation of the Fredholm type:*

$$z(t,\varepsilon) = \int_0^1 \exp\left(\frac{1}{\varepsilon}\int_s^1 \mu(\theta)d\theta\right)G(t,s,\varepsilon)z(s,\varepsilon)ds + \frac{H_1(t,\varepsilon)}{\varepsilon} \tag{20}$$

*where $G(t,s,\varepsilon) = \frac{1}{\varepsilon}\int_0^t e^{\frac{1}{\varepsilon}\int_x^t a(\theta)d\theta}K(x,s)dx$. Let us show that the kernel $G(t,s,\varepsilon)$ of this equation is uniformly bounded for $0 \leq s,t \leq 1$, i.e., which the following statement holds.*

**Lemma 2.** *Let conditions (1) and (2) be satisfied. Then the kernel $G(t,s,\varepsilon)$ is uniformly bounded, i.e., $|G(t,s,\varepsilon)| \leq M$ for all $(s,t,\varepsilon) \in [0,1] \times [0,1] \times (0,+\infty)$.*

**Proof.** Using the operation of integration by parts, we have

$$G(t,s,\varepsilon) = \frac{1}{\varepsilon}\int_0^t e^{\frac{1}{\varepsilon}\int_x^t a(\theta)d\theta}K(x,s)dx = \int_0^t \frac{K(x,s)}{-a(x)}d_x e^{\frac{1}{\varepsilon}\int_x^t a(\theta)d\theta} =$$

$$= \frac{K(x,s)}{-a(x)}e^{\frac{1}{\varepsilon}\int_x^t a(\theta)d\theta}\Big|_{x=0}^{x=t} - \int_0^t e^{\frac{1}{\varepsilon}\int_x^t a(\theta)d\theta}\frac{\partial}{\partial x}\left(\frac{K(x,s)}{-a(x)}\right)dx =$$

$$= \left[\frac{K(t,s)}{-a(t)} - \frac{K(0,s)}{-a(0)} e^{\frac{1}{\varepsilon}\int_0^t a(\theta)d\theta}\right] - \int_0^t e^{\frac{1}{\varepsilon}\int_x^t a(\theta)d\theta} \frac{\partial}{\partial x}\left(\frac{K(x,s)}{-a(x)}\right)dx.$$

Hence, it is clear that under conditions (1) and (2) the kernel $G(t,s,\varepsilon)$ is uniformly bounded, i.e., $|G(t,s,\varepsilon)| \leq M$ for all $0 \leq s, t \leq 1, \varepsilon > 0$. The Lemma 2 is proved.

We now turn to the proof of the correct solvability of Equation (20). To do this, we will try to estimate the norm of the resolvent $R(t,s,\varepsilon)$ of the kernel $\tilde{K}(t,s,\varepsilon) = \exp\left(\frac{1}{\varepsilon}\int_s^1 \mu(\theta)d\theta\right)$ $G(t,s,\varepsilon)$ of integral Equation (20). Let us denote $\chi = \min_{t\in[0,1]} \mathrm{Re}(-\mu(t))$ and estimate the iterated kernels of the integral operator of this system. By Lemma 2, for all $0 \leq s, t \leq 1$ and $\varepsilon > 0$ we have

$$|\tilde{K}_1(t,s,\varepsilon)| \equiv |\tilde{K}(t,s,\varepsilon)| \leq M;$$

$$|\tilde{K}_2(t,s,\varepsilon)| \equiv |\int_0^1 \tilde{K}(t,x,\varepsilon)\tilde{K}_1(x,s,\varepsilon)dx| \equiv$$

$$\equiv |\int_0^1 \exp\left(\frac{1}{\varepsilon}\int_x^1 \mu(\theta)d\theta\right) G(t,x,\varepsilon) \exp\left(\frac{1}{\varepsilon}\int_s^1 \mu(\theta)d\theta\right) G(x,s,\varepsilon)dx| \leq$$

$$\leq M^2 \int_0^1 \exp\left(\frac{1}{\varepsilon}\int_x^1 \mathrm{Re}\,\mu(\theta)d\theta\right)dx \leq M^2 \int_0^1 \exp\left(-\frac{\chi(1-x)}{\varepsilon}\right)dx =$$

$$\leq M^2\varepsilon \frac{\exp\left(-\frac{\chi(1-x)}{\varepsilon}\right)}{x}\Big|_{x=0}^{x=1} = \frac{M^2\varepsilon}{\chi}(1 - e^{\frac{\chi}{\varepsilon}}) \leq \frac{M^2}{\varepsilon},$$

$$|\tilde{K}_3(t,s,\varepsilon)| \equiv |\int_0^1 \tilde{K}(t,x,\varepsilon)\tilde{K}_2(x,s,\varepsilon)dx| \leq \int_0^1 |\tilde{K}(t,s,\varepsilon)| \cdot |\tilde{K}_2(x,s,\varepsilon)|dx \leq$$

$$\leq \frac{M^2}{\chi}\varepsilon \int_0^1 \left(\frac{1}{\varepsilon}\int_x^1 \mathrm{Re}\,\mu(\theta)d\theta\right)|G(t,x,\varepsilon)|dx \leq \frac{M^3}{\chi}\varepsilon \int_0^1 \exp\left(-\frac{\chi(1-x)}{\varepsilon}\right)dx \leq \frac{M^3\varepsilon^2}{\chi^2}.$$

Suppose now that, for $n = r \geq 1$, the estimate

$$|\tilde{K}_r(t,s,\varepsilon)| \leq \frac{M^r\varepsilon^{r-1}}{\chi^{r-1}}, 0 \leq s, t \leq 1, \varepsilon > 0$$

holds. Let us show that this estimate is also true for $n = r + 1$. Indeed,

$$|\tilde{K}_{r+1}(t,s,\varepsilon)| \equiv \int_0^1 |\tilde{K}(t,x,\varepsilon)\tilde{K}_r(x,s,\varepsilon)dx| \leq \int_0^1 |\tilde{K}(t,x,\varepsilon)| \cdot |\tilde{K}_r(x,s,\varepsilon)|dx \leq$$

$$\leq \frac{M^r\varepsilon^{r-1}}{\chi^{r-1}}\int_0^1 |\tilde{K}(t,x,\varepsilon)|dx = \frac{M^{r+1}\varepsilon^{r-1}}{\chi^{r-1}}\frac{\varepsilon}{\chi}e^{-\frac{\chi(1-x)}{\varepsilon}}\Big|_{x=0}^{x=1} =$$

$$= \frac{M^{r+1}\varepsilon^r}{\chi^r}\left(1 - e^{-\frac{\chi}{\varepsilon}}\right) \leq \frac{M^{r+1}\varepsilon^r}{\chi^r} \ (0 \leq s, t \leq 1, \varepsilon > 0).$$

So, for all $0 \leq s, t \leq 1, \varepsilon > 0$ we have proved the estimate

$$|\tilde{K}_n(t,s,\varepsilon)| \leq \frac{M^n\varepsilon^{n-1}}{\chi^{n-1}} (n = 1, 2, 3, \dots).$$

But then the resolvent

$$R(t,s,\varepsilon) \equiv \tilde{K}_1(t,s,\varepsilon) + \tilde{K}_2(t,s,\varepsilon) + \cdots + \tilde{K}_n(t,s,\varepsilon) + \cdots \equiv \sum_{n=1}^{\infty} \tilde{K}_n(t,s,\varepsilon)$$

majorized by a number series

$$\sum_{n=1}^{\infty} \frac{M^n \varepsilon^{n-1}}{\chi^{n-1}} \equiv M \sum_{n=1}^{\infty} \left( \frac{M\varepsilon}{\chi} \right)^{n-1} = \frac{M}{1 - \frac{M\varepsilon}{\chi}}$$

converging absolutely for $0 < \varepsilon < \frac{\chi}{M}$. This means that the series for the resolvent converges absolutely and uniformly in $(s,t) : 0 \le s, t \le 1$ for all $\varepsilon \in (0, \frac{\chi}{2M}]$. In this case, we have the estimate

$$|R(t,s,\varepsilon)| \le \frac{M}{1 - \frac{M\varepsilon}{\chi}} \le 2M,$$

at $(s,t,\varepsilon) : 0 \le s, t \le 1, 0 < \varepsilon \le \varepsilon_0$ (where $\varepsilon_0 > 0$ is small enough). Consequently, for $\varepsilon \in (0, \varepsilon_0]$ Equation (20) (and hence the equivalent Equation $(20_0)$) is uniquely solvable in the class $C^1([0,1], \mathbb{C})$ and its solution is represented in the form

$$z(t,\varepsilon) = \frac{1}{\varepsilon} H_1(t,\varepsilon) + \frac{1}{\varepsilon} \int_0^1 R(t,s,\varepsilon) H_1(t,s,\varepsilon) ds$$

for any right-hand side $H_1(t,\varepsilon) \equiv \int_0^t Y(t,x,\varepsilon) H(x,\varepsilon) dx$. From this we derive the estimate

$$||z(t,\varepsilon)||_{C[0,1]} \le \frac{1}{\varepsilon} ||H_1(t,\varepsilon)||_{C[0,1]} + \frac{1}{\varepsilon} 2M ||H_1(t,\varepsilon)|| \le$$

$$\le \frac{1}{\varepsilon} \left( ||H(t,\varepsilon)||_{C[0,1]} + 2Mc_0 ||H(t,\varepsilon)||_{C[0,1]} \right) \le \bar{c}_0 \frac{||H(t,\varepsilon)||_{C[0,1]}}{\varepsilon}$$

where $\bar{c}_0 = c_0(1 + 2M) > 0$ is a constant independent of $\varepsilon \in (0, \varepsilon_0]$. The following statement is proved. $\square$

**Lemma 3.** *Let conditions (1) and (2) be satisfied. Then, for sufficiently small $\varepsilon (0 < \varepsilon \le \varepsilon_0)$, the Equation $(20_0)$ is uniquely solvable in the class $C^1([0,1], \mathbb{C})$ and its solution satisfies the estimate*

$$||z(t,\varepsilon)||_{C[0,1]} \le \frac{\bar{c}_0}{\varepsilon} ||H(t,\varepsilon)||_{C[0,1]}$$

*where the constant $\bar{c}_0 > 0$ does not depend on $\varepsilon (0 < \varepsilon \le \varepsilon_0]$.*

**Remark 3.** *Correct solvability of the integral system (20) means that the integral operator $\int_0^1 \exp\left( \frac{1}{\varepsilon} \int_s^1 \mu(\theta) d\theta \right) G(t,s,\varepsilon) z(s,\varepsilon) ds$ has no eigenvalues in the space $C([0,1], \mathbb{C})$ (for sufficiently small $\varepsilon > 0$).*

We apply Lemma 3 to prove the following statement.

**Theorem 3.** *Let conditions (1) and (2) be satisfied. Then the problem (1) is uniquely solvable in the class $C^1([0,1], \mathbb{C})$ and its solution $y(t,\varepsilon)$ satisfies the estimate*

$$||y(t,\varepsilon) - y_{\varepsilon N}(t)||_{C[0,1]} \le c_N \varepsilon^{N+1}, N = 0, 1, 2, \dots$$

*where $y_{\varepsilon N}(t)$ is the narrowing (for $\tau = \frac{\psi(t)}{\varepsilon}$), N-th partial sum of the series (6) (with coefficients $y_k(t,\tau) \in U$ satisfying the iterative problems $(9_k)$), and the constant $c_N > 0$ does not depend on $\varepsilon$ at $\varepsilon \in (0, \varepsilon_0] (\varepsilon_0 > 0$ is small enough).*

**Proof.** The problem (1) is uniquely solvable, since it is reduced to the problem $(20_0)$ by a change $y - y^0 = z$. By Lemma 1, for the difference $\Delta_N(t, \varepsilon) = y(t, \varepsilon) - y_{\varepsilon N}(t)$, we obtain the equation

$$\varepsilon \frac{\Delta_N}{dt} = a(t)\Delta_N(t, \varepsilon) + \int_0^1 \exp\left(\frac{1}{\varepsilon}\int_s^1 \mu(\theta)d\theta\right) K(t, s)\Delta_N(s, \varepsilon)ds - \varepsilon^{N+1}F_N(t, \varepsilon), \Delta_N(t, \varepsilon) = 0.$$

It has the form of the problem (20) with inhomogeneity $H(t, \varepsilon) \equiv -\varepsilon^{N+1}F_N(t, \varepsilon)$. By Lemma 3, we have the estimate

$$||\Delta_N(t, \varepsilon)||_{C[0,1]} \equiv ||y(t, \varepsilon) - y_{\varepsilon N}(t)||_{C[0,1]} \leq \frac{\bar{c}_0}{\varepsilon}\varepsilon^{N+1}||F_N(t, \varepsilon)||_{C[0,1]} \leq \bar{c}_0\bar{F}_N\varepsilon^N \equiv \bar{c}_{N-1}\varepsilon^N$$

and, therefore, for $\Delta_{N+1}(t, \varepsilon) = y(t, \varepsilon) - y_{\varepsilon, N+1}(t)$ will have the estimate

$$||\Delta_{N+1}(t, \varepsilon)||_{C[0,1]} \equiv ||(y(t, \varepsilon) - y_{\varepsilon N}(t)) - \varepsilon^{N+1}y_{N+1}(t, \frac{\psi(t)}{\varepsilon})||_{C[0,1]} \leq \bar{c}_N\varepsilon^{N+1}.$$

Hence, we obtain that

$$\bar{c}_N\varepsilon^{N+1} \geq ||y(t, \varepsilon) - y_{\varepsilon N}(t)||_{C[0,1]} - \varepsilon^{N+1}||y_{N+1}(t, \frac{\psi(t)}{\varepsilon})||_{C[0,1]}$$

or $||y(t, \varepsilon) - y_{\varepsilon N}(t)||_{C[0,1]} \leq c_N\varepsilon^{N+1}$, where $c_N = \bar{c}_N + \bar{y}_N > 0, ||y_{N+1}(t, \frac{\psi(t)}{\varepsilon})||_{C[0,1]} \leq \bar{y}_N$, and the constant $c_N$ does not depend on $\varepsilon \in (0, \varepsilon_0]$, where $\varepsilon_0 > 0$ is small enough. The Theorem 3 is proved. □

According to this Theorem 3, the leading term of the asymptotics of the solution the problem (1) has the form (see Formula (16))

$$y_{\varepsilon 0}(t, \sigma) = \frac{h_1(t)}{-\lambda_1(t)} + \alpha_1^{(0)}(t, \sigma)e^{\frac{1}{\varepsilon}\int_0^t a(\theta)d\theta} + \frac{h_2(t)}{\lambda_2(t) - \lambda_1(t)}e^{\frac{i}{\varepsilon}\int_0^t \beta'(\theta)d\theta}\sigma_4 =$$

$$= \frac{h_1(t)}{-\lambda_1(t)} + \left[y^0 + \frac{h_1(0)}{\lambda_1(0)} - \frac{h_2(0)}{\lambda_2(0) - \lambda_1(0)}e^{\frac{i}{\varepsilon}\beta(0)}\right]e^{\frac{1}{\varepsilon}\int_0^t a(\theta)d\theta} + \frac{h_2(t)}{\lambda_2(t) - \lambda_1(t)}e^{\frac{i}{\varepsilon}\beta(t)}. \tag{21}$$

It is clearly seen here how the rapidly oscillating inhomogeneity affects the asymptotic behavior of the solution to Equation (1), but the contribution of the integral operator $\int_0^1 e^{\frac{1}{\varepsilon}\int_s^T \mu(\theta)d\theta}K(t, s)y(s, \varepsilon)ds$ to it is not found; therefore, we calculate the next term of the asymptotics.

Substituting the solution to the problem in the right-hand side, we obtain the following equation:

$$Ly_1(t, \tau, \sigma) = -\frac{\partial y_0}{\partial t} + R_1 y_0 =$$

$$= -\frac{\partial}{\partial t}\left(-\frac{h_1(t)}{\lambda_1(t)} + \alpha_1^{(0)}(t, \sigma)e^{\tau_1} + \frac{h_2(t)}{\lambda_2(t) - \lambda_1(t)}e^{\tau_2}\sigma_4\right) + R_1 y_0 =$$

$$= \left(\frac{h_1(t)}{\lambda_1(t)}\right)^{\bullet} - \dot{\alpha}_1^{(0)}(t, \sigma)e^{\tau_1} - \left(\frac{h_2(t)}{\lambda_2(t) - \lambda_1(t)}\right)^{\bullet}e^{\tau_2}\sigma_4 -$$

$$- \left[\frac{K(t, 1)h_1(1, \sigma)}{\lambda_3(1)\lambda_1(1)}\sigma_3 - \frac{K(t, 0)h_1(0, \sigma)}{\lambda_3(0)\lambda_1(0)}\right] + \left[\frac{K(t, 1)\alpha_1^{(0)}(1, \sigma)}{\lambda_1(1) - \lambda_3(1)}\sigma_1^2 - \frac{K(t, 0)\alpha_1^{(0)}(0, \sigma)}{\lambda_1(0) - \lambda_3(0)}\sigma_3\right] +$$

$$+ \left[\frac{K(t, 1)h_2(1, \sigma)}{[\lambda_2(1) - \lambda_3(1)]^2}\sigma_2^2\sigma_4 - \frac{K(t, 0)h_2(0, \sigma)}{[\lambda_2(0) - \lambda_3(0)]^2}\sigma_3\sigma_4\right] = \left(\frac{h_1(t)}{\lambda_1(t)}\right)^{\bullet} - \dot{\alpha}_1^{(0)}(t, \sigma)e^{\tau_1} -$$

$$- \left(\frac{h_2(t)}{\lambda_2(t) - \lambda_1(t)}\right)^{\bullet}e^{\tau_2}\sigma_4 + \frac{K(t, 0)h_1(0, \sigma)}{\lambda_3(0)\lambda_1(0)} + \frac{K(t, 1)\alpha_1^{(0)}(1, \sigma)}{\lambda_1(1) - \lambda_3(1)}\sigma_1^2 + \frac{K(t, 1)h_2(1, \sigma)}{[\lambda_2(1) - \lambda_3(1)]^2}\sigma_2^2\sigma_4 -$$

$$-\left[\frac{K(t,1)h_1(1,\sigma)}{\lambda_3(1)\lambda_1(1)} + \frac{K(t,0)\alpha_1^{(0)}(0,\sigma)}{\lambda_1(0) - \lambda_3(0)} + \frac{K(t,0)h_2(0,\sigma)}{[\lambda_2(0) - \lambda_3(0)]^2}\sigma_4\right]\sigma_3.$$

Defining the solution of this equation as an element

$$y_1(t,\tau) = y_0^{(1)}(t,\sigma) + \sum_{j=1}^{2} y_j^{(1)}(t,\sigma)e^{\tau_j}$$

of the space $U$, we arrive at the following equality:

$$\sum_{j=1}^{2}\left[\lambda_j(t) - \lambda_1(t)\right]y_j^{(1)}(t,\sigma)e^{\tau_j} - \lambda_1(t)y_0^{(1)}(t,\sigma) =$$

$$= \left(\frac{h_1(t)}{\lambda_1(t)}\right)^{\bullet} - \dot{\alpha}_1^{(0)}(t,\sigma)e^{\tau_1} - \left(\frac{h_2(t)}{\lambda_2(t) - \lambda_1(t)}\right)^{\bullet}e^{\tau_2}\sigma_4 +$$

$$+ \frac{K(t,0)h_1(0,\sigma)}{\lambda_3(0)\lambda_1(0)} + \frac{K(t,1)\alpha_1^{(0)}(1,\sigma)}{\lambda_1(1) - \lambda_3(1)}\sigma_1^2 + \frac{K(t,1)h_2(1,\sigma)}{[\lambda_2(1) - \lambda_3(1)]^2}\sigma_2^2\sigma_4 -$$

$$- \left[\frac{K(t,1)h_1(1,\sigma)}{\lambda_3(1)\lambda_1(1)} + \frac{K(t,0)\alpha_1^{(0)}(0,\sigma)}{\lambda_1(0) - \lambda_3(0)} + \frac{K(t,0)h_2(0,\sigma)}{[\lambda_2(0) - \lambda_3(0)]^2}\sigma_4\right]\sigma_3.$$

Equating here separately the free terms and the coefficients at the exponentials $e^{\tau_j}$, we will have

$$-\lambda_1(t)y_0^{(1)}(t,\sigma) = \left(\frac{h_1(t)}{\lambda_1(t)}\right)^{\bullet} + \frac{K(t,0)h_1(0,\sigma)}{\lambda_3(0)\lambda_1(0)} + \frac{K(t,1)\alpha_1^{(0)}(1,\sigma)}{\lambda_1(1) - \lambda_3(1)}\sigma_1^2 +$$

$$+ \frac{K(t,1)h_2(1,\sigma)}{[\lambda_2(1) - \lambda_3(1)]^2}\sigma_2^2\sigma_4 - \left[\frac{K(t,1)h_1(1,\sigma)}{\lambda_3(1)\lambda_1(1)} + \frac{K(t,0)\alpha_1^{(0)}(0,\sigma)}{\lambda_1(0) - \lambda_3(0)} + \frac{K(t,0)h_2(0,\sigma)}{[\lambda_2(0) - \lambda_3(0)]^2}\sigma_4\right]\sigma_3.$$

$$0 \cdot y_1^{(1)}(t,\sigma) = -\dot{\alpha}_1^{(0)}(t,\sigma),$$

$$[\lambda_2(t) - \lambda_1(t)]y_2^{(1)}(t,\sigma) = -\left(\frac{h_2(t)}{\lambda_2(t) - \lambda_1(t)}\right)^{\bullet}\sigma_4.$$

Since the orthogonality condition $\dot{\alpha}_1^{(0)}(t,\sigma) \equiv 0$ is satisfied, these equations have solutions in the form of functions:

$$y_0^{(1)}(t,\sigma) = -\frac{1}{\lambda_1(t)}\left\{\left(\frac{h_1(t)}{\lambda_1(t)}\right)^{\bullet} + \frac{K(t,0)h_1(0,\sigma)}{\lambda_3(0)\lambda_1(0)} + \frac{K(t,1)\alpha_1^{(0)}(1,\sigma)}{\lambda_1(1) - \lambda_3(1)}\sigma_1^2 +\right.$$

$$\left. + \frac{K(t,1)h_2(1,\sigma)}{[\lambda_2(1) - \lambda_3(1)]^2}\sigma_2^2\sigma_4 - \left[\frac{K(t,1)h_1(1,\sigma)}{\lambda_3(1)\lambda_1(1)} + \frac{K(t,0)\alpha_1^{(0)}(0,\sigma)}{\lambda_1(0) - \lambda_3(0)} + \frac{K(t,0)h_2(0,\sigma)}{[\lambda_2(0) - \lambda_3(0)]^2}\sigma_4\right]\sigma_3\right\},$$

$$y_2^{(1)}(t,\sigma) = -\frac{\left(\frac{h_2(t)}{\lambda_2(t) - \lambda_1(t)}\right)^{\bullet}}{\lambda_2(t) - \lambda_1(t)}\sigma_4$$

and $y_1^{(1)}(t,\sigma) = \alpha_1^{(1)}(t,\sigma) \in C^\infty[0,1]$ is an arbitrary function. Thus, the solution to the problem $(9_1)$ will be as follows:

$$y_1(t,\tau,\sigma) = -\frac{1}{\lambda_1(t)}\left\{\left(\frac{h_1(t)}{\lambda_1(t)}\right)^{\bullet} + \frac{K(t,0)h_1(0,\sigma)}{\lambda_3(0)\lambda_1(0)} + \frac{K(t,1)\alpha_1^{(0)}(1,\sigma)}{\lambda_1(1)-\lambda_3(1)}\sigma_1^2 + \frac{K(t,1)h_2(1,\sigma)}{[\lambda_2(1)-\lambda_3(1)]^2}\sigma_2^2\sigma_4 - \right.$$

$$\left. -\left[\frac{K(t,1)h_1(1,\sigma)}{\lambda_3(1)\lambda_1(1)} + \frac{K(t,0)\alpha_1^{(0)}(0,\sigma)}{\lambda_1(0)-\lambda_3(0)} + \frac{K(t,0)h_2(0,\sigma)}{[\lambda_2(0)-\lambda_3(0)]^2}\sigma_4\right]\sigma_3\right\} + \alpha_1^{(1)}(t,\sigma)e^{\tau_1} - \frac{\left(\frac{h_2(t)}{\lambda_2(t)-\lambda_1(t)}\right)^{\bullet}}{\lambda_2(t)-\lambda_1(t)}e^{\tau_2}\sigma_4$$

where $\alpha_1^{(1)}(t,\sigma) \in C^\infty[0,1]$ is an arbitrary function that is calculated in the process of solving the next iterative problem $(9_2)$. As a result, we obtain an asymptotic solution of the first order:

$$y_{\varepsilon 1}(t) = \frac{h_1(t)}{-\lambda_1(t)} + \left[y^0 + \frac{h_1(0)}{\lambda_1(0)} - \frac{h_2(0)}{\lambda_2(0)-\lambda_1(0)}\sigma_4\right]e^{\frac{1}{\varepsilon}\int_0^t a(\theta)d\theta} + \frac{h_2(t)}{\lambda_2(t)-\lambda_1(t)}e^{\frac{i}{\varepsilon}\int_0^t \beta'(\theta)d\theta}\sigma_4 -$$

$$-\frac{\varepsilon}{\lambda_1(t)}\left\{\left(\frac{h_1(t)}{\lambda_1(t)}\right)^{\bullet} + \frac{K(t,0)h_1(0,\sigma)}{\lambda_3(0)\lambda_1(0)} + \frac{K(t,1)\alpha_1^{(0)}(1,\sigma)}{\lambda_1(1)-\lambda_3(1)}\sigma_1^2 + \frac{K(t,1)h_2(1,\sigma)}{[\lambda_2(1)-\lambda_3(1)]^2}\sigma_2^2\sigma_4 - \right.$$

$$\left. -\left[\frac{K(t,1)h_1(1,\sigma)}{\lambda_3(1)\lambda_1(1)} + \frac{K(t,0)\alpha_1^{(0)}(0,\sigma)}{\lambda_1(0)-\lambda_3(0)} + \frac{K(t,0)h_2(0,\sigma)}{[\lambda_2(0)-\lambda_3(0)]^2}\sigma_4\right]\sigma_3\right\} + \varepsilon\alpha_1^{(1)}(t,\sigma)e^{\frac{1}{\varepsilon}\int_0^t a(\theta)d\theta} -$$

$$-\varepsilon\frac{\left(\frac{h_2(t)}{\lambda_2(t)-\lambda_1(t)}\right)^{\bullet}}{\lambda_2(t)-\lambda_1(t)}\sigma_4 e^{\frac{i}{\varepsilon}\int_0^t \beta'(\theta)d\theta}$$

from which it is seen that the kernel of the integral operator affects only the formation of particular solutions of iterative problems $(9_k)$ and particular solutions of equations for the functions $\alpha_1^{(k)}(t,\sigma)$.

In conditions of solvability of the type (11), as already mentioned above, the integral operator does not participate. This is the main difference between integro-differential equations of Fredholm type from equations of Volterra type, where the kernel of the integral operator significantly affects the construction of the general solution of the equations for functions $\alpha_1^{(k)}(t,\sigma)$ (see, for example, [20]).

## 6. Conclusions

Since the terms of order $\varepsilon$ in $y_{\varepsilon 1}(t)$ uniformly tend to zero, when $\varepsilon \to +0$, then the behavior of the exact solution of the problem (1) as the small parameter tends to zero completely is determined by its main term of asymptotics (21): after leaving the point $y = y^0$ at $t = 0$, the exact solution $y(t,\varepsilon)$ of the problem (1) (for $t > 0$ and $\varepsilon \to +0$) will perform fast oscillations around the "degenerate solution" $\bar{y}(t) = \frac{h_1(t)}{-\lambda_1(t)}$, not tending for any limit.

**Author Contributions:** All authors contributed evenly. All authors have read and agreed to the published version of the manuscript.

**Funding:** This work was supported by grant No. AP05133858 of the Ministry of Education and Science of the Republic of Kazakhstan.

**Institutional Review Board Statement:** Not applicable.

**Informed Consent Statement:** Not applicable.

**Data Availability Statement:** Not applicable.

**Conflicts of Interest:** The funders had no role in the design of the study; in the collection, analyses, or interpretation of data; in the writing of the manuscript, or in the decision to publish the results.

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
