# Peer review of "Regularized Asymptotic Solutions of a Singularly Perturbed Fredholm Equation with a Rapidly Varying Kernel and a Rapidly Oscillating Inhomogeneity"

_axioms, doi:10.3390/axioms11030141_

Round 1

Reviewer 1 Report

This paper deals with a regularized asymptotic solution of a singularly perturbed Fredholm equation with a rapidly varying kernel and a rapidly oscillating inhomogeneity. I would like to point out or comment following as:

  1. No abstract, is that right?
  2. In the introduction, the description of the object in this paper is very ambiguous.
  3. What is the differentiation compared to previous works? What is novelty? It is not clear. I cannot find the novelty. Please add it.
  4. What is the main purpose of this paper?
  5. I hope the authors should add what is an application for the proposed method.
  6. Can you show the result by the figure which when authors used by computer?
  7. I hope the authors should describe it logically.

Author Response

Response to Reviewer 1 Comments

  1. No abstract, is that right?

Answer: No. We included abstract.

  1. In the introduction, the description of the object in this paper is very ambiguous.

Answer: You're right. We have changed the introduction by writing down specifically the problem (2), which we will consider in our article.

  1. What is the differentiation compared to previous works? What is novelty? It is not clear. I cannot find the novelty. Please add it.

Answer: The novelty is included in the fact that the Fredholm equation (2) contains rapidly oscillating terms in the right part. Such problems have not been considered before.

  1. What is the main purpose of this paper?

Answer: The main purpose of the work is to generalize the Lomov’s regularization method to the problems of type (2) and to identify how the rapidly oscillating inhomogeneity affects the asymptotics of the solution.

  1. I hope the authors should add what is an application for the proposed method.

Answer: Such problems may arise in electrical engineering, electronics, etc., where rapidly oscillating external forces are involved.

  1. Can you show the result by the figure which when authors used by computer?

Answer: Numerical calculation of the asymptotics has not been carried out, so there are no figures in the work.

  1. I hope the authors should describe it logically.

Answer: We have responded to all the reviewer's comments in detail and logically. In the paper, all the results are strictly mathematically justified.

Reviewer 2 Report

There is no any novelty in this submission. Authors repeated the results previously derived. I would suggest to reject this paper with an encouragement to resubmit the new version with clear statement of the novelty comparing with previous results including results of the last author (Prof. V.F. Safanov). Fore sake of readers examples are also expected.

Author Response

Response to Reviewer 2 Comments

Answer: 

In the responses of the first reviewer, as well as in the work itself, it was pointed out that the problem considered in the article had not previously been studied from the point of view of the Lomov’s regularization method. The novelty lies in the fact that we study a problem with an integral operator of Fredholm type with rapidly oscillating inhomogeneity.  Earlier, the works of Safonov and other authors investigated integro-differential equations of the Volterra type and with slowly varying inhomogeneities.

Reviewer 3 Report

The paper deals with the study of the integro-differential equation and its initial condition (1) equipped with rapidly varying kernel function. The Authors demonstrate a procedure for the construction of an asymptotic solution of the initial-value problem (1) (IVP). However they assumed that α=T=1. Thereafter the IVP in question is investigated with two important conditions (see the beginning of Section 2). For solvability the Authors present and prove Theorem 1 as necessary and sufficient condition and Theorem 2 for the uniqueness of the solution (see Section 3). The asymptotic convergence analysis is carried out in Section 5 (see Lemma 1, 2 and 3 and Theorem 3).

The paper is well-structured and well-written in English. The presented definitions, lemmas, theorems and proofs elaborated are clear, understandable and thorough enough, do not require any modifications and explanations. However,

  • it would be worth demonstrating the main results (lemmas and theorems above-mentioned) through the illustration of a representative model problem, thereby giving as an example for the solution procedure as well.
  • Section 'Abstract' should be inserted in the text as well.
  • In Section Conclusions, a good summary about the paper and mainly the achieved results should be inserted. To the reviewer's opinion, the important statements described in 'Conclusions' belong rather to the end of the main term of the paper. 

To sum up: this is an apposite, interesting and clearly written paper which can be recommended for publication in Axioms, but, at least, a minor revision is needed (see the details above).

Author Response

Response to Reviewer 3 Comments

Answer:

This reviewer analyzed our work with understanding. We are grateful to him for the work done and the positive feedback on it.

Round 2

Reviewer 1 Report

This paper is well revised according to reviewer's point out. Thus, I would like to decided ac an "accept"

Author Response

Responses to reviewer's comments No. 1.

  1. No abstract, is that right?

Answer: No. We included abstract.

  1. In the introduction, the description of the object in this paper is very ambiguous.

Answer: You're right. We have changed the introduction by writing down specifically the problem (2), which we will consider in our article.

  1. What is the differentiation compared to previous works? What is novelty? It is not clear. I cannot find the novelty. Please add it.

Answer: The novelty is included in the fact that the Fredholm equation (2) contains rapidly oscillating terms in the right part. Such problems have not been considered before.

  1. What is the main purpose of this paper?

Answer: The main purpose of the work is to generalize the Lomov’s regularization method to the problems of type (2) and to identify how the rapidly oscillating inhomogeneity affects the asymptotics of the solution.

  1. I hope the authors should add what is an application for the proposed method.

Answer: Such problems may arise in electrical engineering, electronics, etc., where rapidly oscillating external forces are involved.

  1. Can you show the result by the figure which when authors used by computer?

Answer: Numerical calculation of the asymptotics has not been carried out, so there are no figures in the work.

  1. I hope the authors should describe it logically.

Answer: We have responded to all the reviewer's comments in detail and logically. In the paper, all the results are strictly mathematically justified.

Reviewer 2 Report

There were no changes to demonstrate the novelty of the manuscript comparing with authors and coauthors previous works. The technique used is similar for both rapidly oscillating inhomogeneity and slowly varying inhomogeneities. Please outline the novelty more clear for sake of readers. The article is very hard to read. Please use a latex style file with continuous line numbering.

Author Response

Responses to reviewer's comments No. 2.

Answer: 

In the responses of the first reviewer, as well as in the work itself, it was pointed out that the problem considered in the article had not previously been studied from the point of view of the Lomov’s regularization method. The novelty lies in the fact that we study a problem with an integral operator of Fredholm type with rapidly oscillating inhomogeneity.  Earlier, the works of Safonov and other authors investigated integro-differential equations of the Volterra type and with slowly varying inhomogeneities.
